# Fulvic Acid Attenuates Atopic Dermatitis by Downregulating CCL17/22

**DOI:** 10.3390/molecules28083507

**Published:** 2023-04-16

**Authors:** Chenxi Wu, Anqi Lyu, Shijun Shan

**Affiliations:** 1Department of Dermatology, Xiang’an Hospital of Xiamen University, School of Medicine, Xiamen University, Xiamen 361000, China; 2Hangzhou Third People’s Hospital, Affiliated Hangzhou Dermatology Hospital, Zhejiang University School of Medicine, Hangzhou 310009, China; 3Chen Hongduo Academician Workstation, Shaoxing Central Hospital, Shaoxing 312030, China

**Keywords:** atopic dermatitis (AD), fulvic acid (FA), CCL17, CCL22

## Abstract

The main pathogenic factor in atopic dermatitis (AD) is Th2 inflammation, and levels of serum CCL17 and CCL22 are related to severity in AD patients. Fulvic acid (FA) is a kind of natural humic acid with anti-inflammatory, antibacterial, and immunomodulatory effects. Our experiments demonstrated the therapeutic effect of FA on AD mice and revealed some potential mechanisms. FA was shown to reduce TARC/CCL17 and MDC/CCL22 expression in HaCaT cells stimulated by TNF-α and IFN-γ. The inhibitors showed that FA inhibits CCL17 and CCL22 production by deactivating the p38 MAPK and JNK pathways. After 2,4-dinitrochlorobenzene (DNCB) induction in mice with atopic dermatitis, FA effectively reduced the symptoms and serum levels of CCL17 and CCL22. In conclusion, topical FA attenuated AD via downregulation of CCL17 and CCL22, via inhibition of P38 MAPK and JNK phosphorylation, and FA is a potential therapeutic agent for AD.

## 1. Introduction

Atopic dermatitis, this chronic skin disease is characterized by pruritus, eczematous skin damage, and a number of clinical symptoms. It is triggered by a defective immune response. AD tends to occur in infants and children, but it can also occur in adults. Although AD can occur at any age, it usually develops in early childhood, such as in toddlers between 3 and 6 months of age [1]. Patients’ families are also significantly affected, and AD also puts a large economic and social burden on patients’ families. Compared with healthy people, children with moderate and severe AD have restless sleep, increased wakefulness, and difficulty falling asleep. The night care and comforting of children also reduces the sleep time of parents by about 1–3 h every night [2]. AD patients also often suffer from poor concentration, due to lack of sleep, sedative medications, and unbearable itching, which can negatively affect future academic performance and career prospects. The patient’s comfort and mental health are both affected by the condition [3,4]. Filaggrin, loricrin, involucrin, and total ceramide levels are all lowered in AD skin lesions, along with other key skin barrier-related proteins. Skin Th2 inflammation is a key part of the pathogenesis of AD. Although there are multiple pathways to the pathogenesis of various AD agents, Th2 immunoinflammation remains a major focus regarding the pathogenesis and development of treatments for AD. The pathological features of AD patients’ skin lesions show T-cell infiltration, dominated by CD4, followed by activation of various inflammatory cells. To attract immune cells to inflamed skin, keratinocytes release chemokines and cytokines after the epidermal barrier is damaged. Lymphodynamics and activation of cells are controlled by chemokines, including TSLP, IL-25, IL-33, TARC/CCL17, and MDC/CCL22 [5]. T-helper (Th2) cells trigger immunological responses associated with Th2 through expressing IL-4 and IL-13.

Keratinocytes release CCL17 and CCL22, which are members of the CC chemokine family [6]. The induction of migration of Th2 CD4+ T cells by CCL17 and 22 chemokines is specific. CCL17 and CCL22 bind to the CCR4 ligand on the surface of Th2 immune cells [7], attracting Th2 cells to infiltrate the lesion and aggravate the disease [8,9]. CCL17 and CCL22 levels in AD patients are associated with disease severity. The treatment of Th2 inflammation-induced skin illnesses such as AD might be possible by inhibiting the CCL17 and CCL22 synthesis in keratinocytes. The MAPK signaling pathway is closely related to AD, and several drugs have been shown to treat AD by regulating this signaling pathway. MAPKs are serine/threonine protein kinases that can be activated by a variety of extracellular stimuli, including growth factors, cytokines, environmental factors, and oxidative stress, and also regulate cell response to external stress by transferring extracellular signals to cells. Classical mammalian MAPKs include the c-Jun amino terminal kinase (JNK1/2/3), P38, and extracellular signal-regulated kinase (ERK) families [10]. The MAPK pathway is involved in a variety of cellular activities, such as gene expression, cell proliferation and metabolism, apoptosis, cell motor regulation, and mitosis. Once this signaling pathway is disrupted, this can lead to many diseases, such as cancer, inflammatory skin disease AD, and so on.

The Ming Dynasty’s “Compendium of Materia Medicine” documented a dark gold stone (composed of fulvic acid) with a sweet and pungent taste that was used for knife cuts, unintentionally ingesting gold and silver, and profuse menstrual flow. In addition to its anti-inflammatory, antiulcer, hemostatic, blood-activating, and antibacterial properties, fulvic acid (FA) also has antiviral and antibacterial properties. It has been applied in many fields, with remarkable effects, and no serious adverse reactions have been reported for the use of fulvic acid, showing a high safety level [11,12,13,14]. We tested fulvic acid’s effect on BALB/c mice by stimulating them with DNCB (2,4-dinitrochlorobenzene). TNF-α and INF-γ were used to stimulate keratinocyte growth in vitro, as a means of investigating the mechanisms underlying the effect. Based on the results of this study, the treatment of atopic dermatitis with FA is a viable therapeutic option.

## 2. Results

### 2.1. Human Keratinocyte (HaCaT) Cell Viability after Exposure to Various FA Concentrations

In order to evaluate the effect of fulvic acid (FA) on HaCaT cell viability, CCK-8 was used to detect the effect of 10–1000 g/mL FA on HaCaT cell viability. Up to 900 g/mL, there was no evidence of cytotoxicity, as shown in Figure 1. As a result, we focused on FA treatment at concentrations of 200 and 500 g/mL.

### 2.2. FA Reduced Pro-Inflammatory Cytokine Expression in TNF-α/IFN-γ-Stimulated HaCaT Cells

TNF-α/IFN-γ, which is extensively employed in atopic dermatitis simulations, acts on HaCaT cells and induces them to release chemokines and cytokines, such as IL-6, IL-8, CCL17, and CCL22 [15,16,17]. CCL17 and CCL22 are essential targets for therapy of AD [6,18]. FA caused a substantial drop in CCL17 and CCL22 mRNA expression (Figure 2A). FA was found to inhibit growth at concentrations up to 500 mg/mL, with a dose-dependent effect. The cells were then pretreated with FA for 30 min, before being exposed to TNF-α/IFN-γ for 20 h. Additionally, the impact of FA on the mRNA levels of cytokines and chemokines was evaluated using RT-qPCR. The mRNA expression levels of TNF-α, IL-6, CCL2, CCL17, and CCL22 were observed at 200 and 500 g/mL FA compared with the AD model group (Figure 2). We also used cellular immunofluorescence staining (ICC) to evaluate the expression of CCL17 and CCL22 in HaCaT cells. According to the immunocytochemistry (ICC) data, FA treatment significantly decreased the HaCaT cell production of CCL17 and CCL22 (Figure 2B). FA was observed to lower the amount of CCL17 and CCL22 protein expression in HaCaT cells boosted with TNF-α/IFN-γ. 

### 2.3. Activated HaCaT Cells Were Inhibited by FA, through Inhibiting the MAPK Pathway

The epidermal keratinocyte activates various signaling pathways, including JNK, p38 MAPK, and ERK [12]. The MAPK phosphorylation pathway has been shown to reduce the symptoms associated with AD skin inflammation, since it produces inflammatory mediators and decreases Th2 cell activity [19,20]. Therefore, we further investigated how FA affects MAPK signaling. Using Western blotting, we analyzed the phosphorylation levels of p38, JNK, and ERK. The model group triggered by TNF-α/IFN-γ showed a considerable increase in JNK and p38 phosphorylation. Glucocorticoids are commonly used for treating AD, for example Dexamethasone (Dex). The positive control group consisted of 10 milligrams of Dex. Positive controls significantly inhibited p38 phosphorylation compared to the model controls. The densities of bands were determined. In the FA-treated HaCaT cell group, 200 μg/mL FA slightly inhibited the p38 and JNK phosphorylation; 500 μg/mL FA significantly inhibited phosphorylation, and the inhibition was dose-dependent (Figure 3).

### 2.4. CCL17 and CCL22 Expression in HaCaT Cells Is Regulated by p38 MAPK and JNK

On the basis of our data, we investigated the relationship between p38 MAPK and JNK kinases in HaCaT cells stimulated by TNF-a/IFN-γ expressing CCL17 and CCL22. SP600125 is a JNK inhibitor, while SB202190 is a p38 inhibitor. As a result, we explored how CCL17 and CCL22 were regulated by p38 MAPK and JNK kinases, using SB202190 and SP600125 (Figure 4A, B). For 30 min after incubation with SB202190 or SP600125, we activated HaCaT cells with TNF-α and IFN-γ. SB202190 administration resulted in a substantial drop in CCL17 and CCL22 mRNA expression. CCL17 mRNA expression was inhibited just as much by SB202190 as by the AD group (Figure 4A). The inhibitory effects of SP600125 were slightly weaker than those of SB202190, in reducing CCL22 mRNA levels and inhibiting CCL17 and CCL22 mRNA production (Figure 4B). TNF-α and IFN-γ-activated HaCaT cells suppressed the production of CCL17 and CCL22 mRNA via the p38 MAPK pathway.

### 2.5. BALB/c Mice Treated with FA Show Fewer Symptoms of AD

Mice ears were repeatedly exposed to DNCB to cause AD-like skin damage, in order to evaluate the results of our in vitro experiments [21]. For the in vivo experiments, FA concentrations of 1 mg/mL and 5 mg/mL were used. We dissolved 1% DNCB in 3:1 acetone/olive oil and applied this to the mice ears. FA was dissolved in deionized water to the indicated concentration. We used Dex as the positive control, and the concentration of Dex was 0.2 mg/kg [22,23]; deionized water was used as the vehicle group for the negative control. On day one of the study, each ear received 20 uL of 1% DNCB, except for the controls. On the following days, each ear was treated every three days with 20 μL of 0.5% DNCB. FA was topically treated every three days.

We measured ear thickness on day 14 and recorded changes in the skin lesions on the ear. It is evident from Figure 5B that repeated use of DNCB caused AD-type skin damage, manifested by erythema, edema, and ear moss growth. The DNCB-treated group had an increase in ear thickness on day 4, followed by the skin thickening, serious redness, inflammation, and desquamation. Initially, we observed a significant improvement in redness and desquamation of the auricle lesions after topical administration of FA (Figure 5B). From about the tenth day, except for DNCB group and vehicle group, the ear thickness of mice began to decrease gradually. Moreover, it was found that FA treatment could significantly reduce the ear thickness and swelling; at the last measurement, ear thickening was significantly reduced in the FA group, especially at higher doses (Figure 5C). In the subsequent treatment, the difference was noted to be more remarkable compared with the control and model groups. A dose-dependent therapeutic effect was observed with high-dose FA compared to low-dose FA. Histopathology was measured using H&E staining on the ear tissues (Figure 5B). There were noticeable changes in the AD animal models, including macrophage infiltration and epidermal hyperplasia [24]. In the DNCB-induced AD model, the epidermal and dermal thicknesses increased significantly. There was a significant reduction in epidermal and dermal thickness after the FA treatment (Figure 5D–E). The thickness of the epidermal and dermal skin was reduced in lesions induced by DNCB when FA was applied.

Low-dose FA treatment attenuated the thickness of ears by 55.32 %, and high-dose FA treatment attenuated the thickness of the ear epidermis by 62.49%, which alleviated the pathological changes to the ear histopathology. Dex had the most obvious effect on reducing ear epidermal thickening (77.82%). Compared with the DNCB-induced mice models, low-dose FA treatment reduced the thickness of ear skin by 24.07%, while high-dose FA treatment reduced the thickness of ear skin by 37.06% (Figure 5B–E). In vivo, we demonstrated that FA could inhibit the manifestation and pathological progression of AD.

### 2.6. FA Substantially Reduced the CCL17 and CCL22 Level of Serum in DNCB-Induced BALB/C Mice

It was shown in vitro that FA reduced the levels of mRNA for CCL17 and CCL22. After FA therapy, serum CCL17 and CCL22 levels were detected using ELISA. The CCL17 and CCL22 levels were considerably reduced by FA therapy compared to the model group (Figure 6). Model groups induced with DNCB were not substantially different from the vehicle groups or low-dose FA groups in terms of serum levels of CCL17 and CCL22. The high-dose FA significantly reduced the CCL17 and CCL22 amounts in the circulation when compared to those of the low-dose FA and vehicle groups. These above results demonstrated that the high-dose FA treatment effectively decreased the serum CCL17 and CCL22 and exerted an anti-inflammatory effect.

## 3. Discussion

Atopic dermatitis (AD) is a complex chronic pruritus inflammatory skin disease caused by Th2 inflammation. Skin Th2 inflammation is a key part of the pathogenesis of AD. Although there are multiple pathways to the pathogenesis of various AD agents, Th2 immunoinflammation remains a major focus regarding the pathogenesis and treatment of AD. The pathological features of AD patient skin lesions showed T-cell infiltration dominated by CD4, followed by activation of various inflammatory cells. CD4 T lymphocytes differentiate into Th2 lineages, and activated Th2 cells release IL-4 and IL-13, which promote the production of antigen-specific IgE and eosinophils by B cells through signal transduction and the transcriptional activator pathways [25]. Atopic dermatitis patients may experience an overexpression of inflammatory factors if their keratinocytes are activated [16,20]. TNF-α/IFN- γ are commonly applied in the modeling of AD cell models [26,27]. We used this model as a model for in vitro experiments. As soon as keratinocytes are activated, they produce cytokines and chemokines, which act as anti-inflammatory agents. A damaged epidermal barrier stimulates keratinocytes, leading to the release of many cytokines and chemokines [28]. We found that FA could significantly inhibit the mRNA expression levels of TNF-α, IL-6, CCL2, CCL17, and CCL22 in HaCaT cells stimulated by TNF-α/IFN-γ. Among them, CCL17 is a chemokine of activation regulation produced by the thymus, and CCL22 is a chemokine derived from giant phage, leading the Th2-led immune response [29]. There is a substantial relationship between the amount of CCL17 and CCL22 in the circulation and the severity of AD [6,18]. Th2 immune cells infiltrate the lesion and aggravate the condition by binding to CCR4 ligands on the surface of CCL17 and CCL22 [8,9]. They can irritate the skin and cause allergies, and may enhance the immune response. There is a lot of evidence that these two chemokines are related to AD severity [28,30]. Thus, we investigated CCL17 and CCL22 inhibition by FA.

MAPK is considered a key regulator of inflammatory diseases [31]. Our experiments showed that FA treatment inhibited TNF-α/IFN-γ-induced CCL17 and CCL22 production by affecting the P38 MAPK and JNK pathways, as evidenced by the use of inhibitors; FA treatment inhibited TNF-α/IFN-γ-induced CCL17 and CCL22 production. Furthermore, FA treatment greatly reduced AD manifestations, and we confirmed that the activation of CCL17 and CCL22 was effectively inhibited in AD-like mouse models induced by DNCB.

To assess the efficacy of FA in vivo, FA was used in DNCB-induced AD models [32,33]. Skin manifestations such as AD are caused by repeated allergic reactions and inflammation. DNCB’s repeated triggering of allergic inflammation in mice resulted in redness, swelling, and lichenization of the skin surface, resulting in epidermal thickening, rupture, bleeding, and scabs. Our experiments demonstrated that topical FA treatment reduced the severity of skin lesions, suppressed immune response, improved histological features, and alleviated epidermal and dermal thickening in mice. Interestingly, FA inhibited cytokine production in Th2 cells, which may reduce the symptoms of AD. A dose-dependent reduction in serum CCL17 and CCL22 levels was also observed following FA treatment.

In conclusion, we found that FA inhibits TNF-α/ IFN-γ-induced CCL17 and CCL22 expression by regulating the P38 MAPK and JNK signaling pathways. Our results suggest that FA may be a potential therapeutic agent for AD. Further preclinical and clinical studies are needed to verify the anti-inflammatory activity of FA. In the future, we will further explore the application of FA in different inflammatory and allergic skin diseases, to determine whether it can relieve and inhibit other pathological symptoms.

## 4. Material and Methods

### 4.1. Cell

Human keratinocytes (HaCaT cells) were used in this paper, which are adherent cells. The cell culture was performed in MEM complete medium (with 10% FBS and 1% penicillin-streptomycin), and CO_2_ incubation at 37 °C, 5%. Purchased from the National Infrastructure of Cell Line Resource, China.

### 4.2. Animals

BALB/C mice (eight weeks old) were provided by Beijing Vital River Laboratory Animal Technology Co., Ltd. (Beijing, China). Laboratory animals were maintained on a 12 h light/dark cycle. The temperature of the laboratory was 18 °C and the humidity was 30%. Drinking and eating were allowed for all mice. Animal care and use committee approval was obtained from Xiamen University (XMULAC20190070).

### 4.3. CCK-8 Assay

Ninety-six-well culture plates were used to seed the HaCaT cells. In each well, 10 μL of CCK8 agent was added after FA treatment for 24 h. An incubator at 37 °C was used to keep the cells from 40 min to 1.5 h. We calculated the cell viability using absorbance at 450 nm and 450 nm.

### 4.4. RT-PCR (Real-Time Quantitative PCR)

Approximately six wells were used for the culturing of HaCaT cells. After discarding the supernatant, 750 μL of Trizol (Sigma USASt. Louis, MI, USA) was added, followed by sample collection and storage at −80 °C or direct RNA extraction. Cells were extracted with an RNeasy Mini Kit (Qiagen Hilden, Hilden, Germany). All operations were carried out in strict accordance with instructions. A Superscript III RT-PCR Kit (Invitrogen, Waltham, MA, USA) was used to convert RNA into cDNA. cDNA synthesis was performed using a C1000 Touch™ Thermal Cycler (Bio-Rad, Hercules, CA, USA). Real-time polymerase chain reaction (PCR) used a real-time polymerase chain reaction system (qTower^3^; Analytik Jena, Jena, Germany) in a 20 μL mixture, containing 10 μL SYBR Green fluorescence (Bimake, Houston, TX, USA) 5 μL primer, and 5 μL quantified cDNA template. TBP was used to normalize the expression. The primer sequences were as follows: hTBP, forward, 5′-ATG ATG CCT TAC GGC ACA GG-3′ and reverse, 5′-GTT GCT GAG ATG TTG ATT GC TG-3′; hIL-6, forward, 5′-CCA TCT TTG GAA GGT TCA GGT TG-3′ and reverse, 5′-GGA CCG AAG GCG CTT GTG GAG-3′; hTNF-α, forward, 5′-ATG GGC TCC CTC TCA TCA GT-3′ and reverse, 5′-GAA ATG GCA AAT CGG CTG AC-3′; hCCL17, forward, 5′-GTC TTG AAG CCT CCT CAC CC-3′and reverse, 5′-GGA TCT CCC TCA CTG TGG CT-3′; hCCL22, forward, 5-ATC GCC TAC AGA CTG CAC TC-3′ and reverse, 5′-GAC GGT AAC GGA CGT AAT CAC-3′; hCCL2, forward, 5′-CAG CCA GAT GCA ATC AAT GCC-3′and reverse, 5′-TGG AAT CCT GAA CCC ACT TCT-3′.

### 4.5. Western Blot Assay

We isolated the total proteins from cells using RIPA buffer (Genstar, Beijing, China) supplemented with protease inhibitors (Genstar, Beijing, China). To quantify the protein isolated from cells and tissues, a BCA Protein Assay Kit (Abcam UK, Cambridge, UK) was employed. Sample loading buffer and lysis buffer contained equal amounts of protein. After boiling at 90 °C for 3 min, proteins were run on Future PAGE gels (Future CHN) for 20 min at 160 volts. After the proteins were transferred to the polyvinylidene fluoride membrane (Invitrogen), BSA-TBST Western Blocking Buffer (Boster CHN) was used to block the membrane at room temperature, for about one and a half hours. Anti-p-JNK antibodies, anti-JNK antibodies, anti-p-p38 MAPK antibodies, and anti-p38 MAPK antibodies (Cell Signaling Technology USA, Danvers, MA, USA) were incubated overnight at 4 °C. Monoclonal antibodies against tubulin (Abclonal CHN) were also used. The secondary antibody was incubated with the membrane at room temperature for one hour. Donkey anti-mouse and a rabbit anti-mouse antibodies were used as secondary antibodies, from LI-COR USA. An infrared imaging system from LI-COR USA (LI-COR Odyssey) was used to visualize the protein bands.

### 4.6. Animals and Treatment

1-chloro-2,4-dini-trochlorobenzene (DNCB, Sigma USA) was dissolved in the vehicle, which was a mixture of acetone and olive (3:1 ratio). Then, 100 μL 1% DNCB was applied to the abdomen on the first day. On Days 4, 7, 10, and 1, 0.5% DNCB 10 μL was applied to each ear of the mice. We split the BALB/c mice into six groups (*n* = 5) at random: (1) A control group (normal control, CTRL) was untreated; (2) DNCB sensitized and challenged group (DNCB); (3–4) FA 1 mg/mL and 5 mg/mL mixed with vehicle were used for sensitized and challenged groups (groups FA 1 and FA 5). (5) Dexamethasone group as a positive drug control (DEX); (6) vehicle group as placebo, sensitized with DNCB and treated with deionized water (Vehicle). CO2 euthanization was performed on day 15. Fulvic acid (CAS: 479-66-3) was obtained from MACKLIN. The chemical structure of FA is illustrated in Figure 7.

### 4.7. Histological Analysis

A 4% formalin fixation was used on all skin tissue specimens for 24 h at room temperature. A perpendicular cut was made in the back skin sections after fixation in graded ethanol, xylene, and paraffin. After staining the sections with hematoxylin and eosin (H&E), they were examined under a microscope. A slide scanner (Motic VM1) was used to evaluate all slides for histology. Four tissues were randomly selected for epidermal and dermal thickness analysis after staining. Image J was used to measure the epidermis and dermis thickness.

### 4.8. ICC (Immunocytochemistry)

Twelve-well plates were seeded with HaCaT cells. Experimental procedures were carried out as described above. After a PBS rinsing and a 20 min fixation with 4% paraformaldehyde at room temperature, the cells were fixed for another 20 min at room temperature. Cells were permeabilized in 0.1% Triton X-100 through PBS for 15 min after fixation. After an hour, the cells were blocked using PBS containing 5% BSA. The cells were treated with a diluted primary antibody overnight at 4 °C. The material was colored with DAPI for 10 min at 37 °C after two hours of incubation with the secondary antibody. Analysis of all samples was performed using a Nikon D-Eclipse confocal laser scanning microscope (Tokyo, Japan). AffiniPure Donkey anti-rabbit or anti-mouse IgG (H + L) and Alexa Fluor^®^ 488/Cy3 antibodies (Jackson Immuno Research USA, West Grove, PA, USA) were used to detect anti-CCL17 and anti-CCL22 antibodies.

### 4.9. ELISA (Enzyme-Linked Immunosorbent Assay)

Samples of mouse serum were tested using ELISA kits, to determine the CCL17 and CCL22 levels. The kits are made by R&D Systems (Emeryville, CA, USA).

### 4.10. Statistical Analysis

There were three independent replications of each experiment. They were analyzed using the statistical software Prism 9 (GraphPad, San Diego, CA, USA), using means ± SEM. In order to determine whether there were significant differences between groups, a *t*-test or one-way ANOVA paired with Dunnett’s multiple comparison test was used. Differences were considered significant if the *p* value was greater than 0.5.

## 5. Conclusions

Our results suggest that FA can reduce the expression levels of CCL17 and CCL22 in HaCaT by inhibiting the phosphorylation of p38 MAPK and JNK kinases in the MAPK signaling pathway. In addition, FA treatment significantly alleviated AD-like lesions in mice in vivo, which verified the inhibitory effect of FA on the expression of CCL17 and CCL22. These results suggest that FA can inhibit the levels of CCL17 and CCL22 in AD to treat AD. FA may be a potential drug for the treatment of AD, which needs to be verified by further experiments.

## Figures and Tables

**Figure 1 molecules-28-03507-f001:**
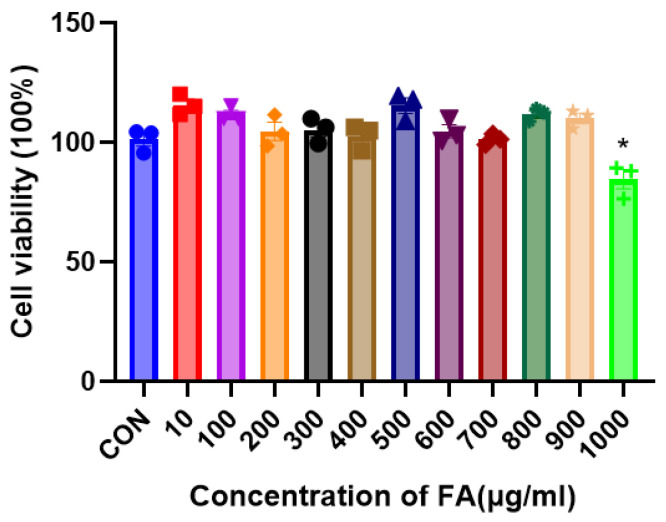
HaCaT cell viability after pre-treatment with various FA doses (10–1000 g/mL). HaCaT cells received different FA doses for 24 h at 37 °C and 5% CO_2_. A minimum of three experiments were used to calculate the mean ± SEM. Compared with the CON group, * *p* < 0.05, ns stands for not significant.

**Figure 2 molecules-28-03507-f002:**
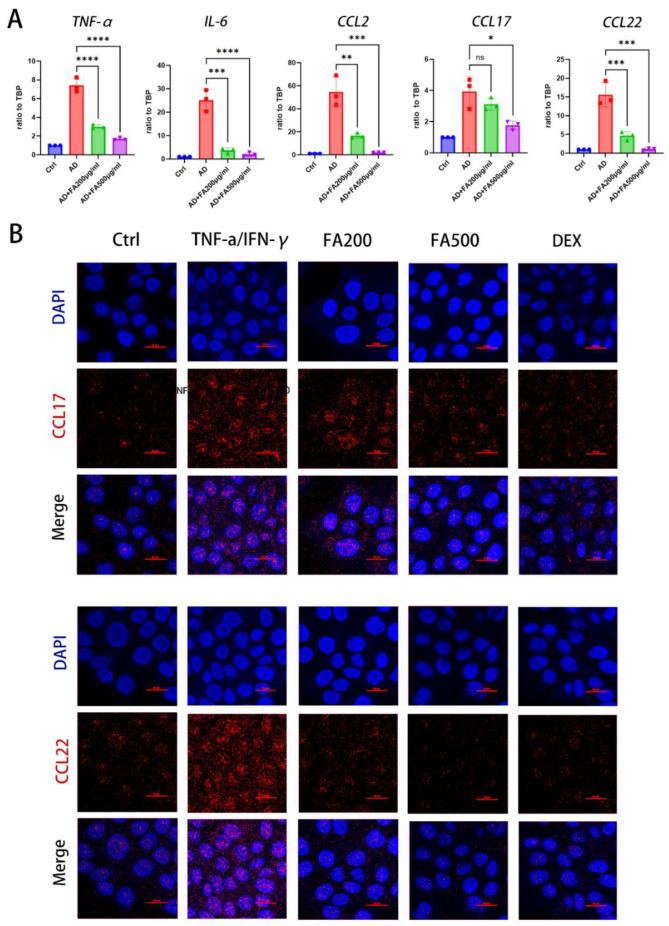
Effects of FA on cytokine mRNA expression in TNF-α/IFN-γ-stimulated HaCaT cells. FA (200 and 500 g/mL) was applied to the cells for 30 min, followed by TNF-α/IFN-γ (10 ng/mL) for 20 h (qRT-PCR) or 30 min (ICC). (**A**) The levels of mRNA expression were evaluated using RT-PCR after extracting the total RNA. (**B**) After 30 min of stimulation, the protein levels of CCL17 and CCL22 were assessed using the intensity of the ICC stain. The positive control was dexamethasone (Dex, a glucocorticoid used for AD). The image scale is 100 um. A minimum of three experiments were used to calculate the mean ± SEM. **** *p* < 0.0001, *** *p* < 0.001, ** *p* < 0.01, * *p* < 0.05, ns stands for not significant.

**Figure 3 molecules-28-03507-f003:**
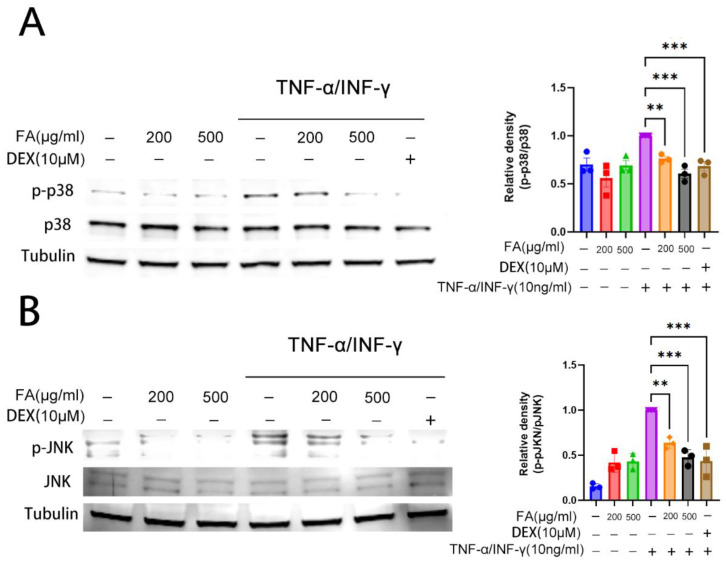
Effects of FA on the p38 MAPK and JNK pathways. TNF-a/IFN-γ (10 ng/mL) was added after pre-treatment with FA (200 and 500 g/mL). (**A**) p38 MAPK, (**B**) JNK Western blot and quantitative analysis. A minimum of three experiments were used to calculate the mean ± SEM., *** *p* < 0.001, ** *p* < 0.01.

**Figure 4 molecules-28-03507-f004:**
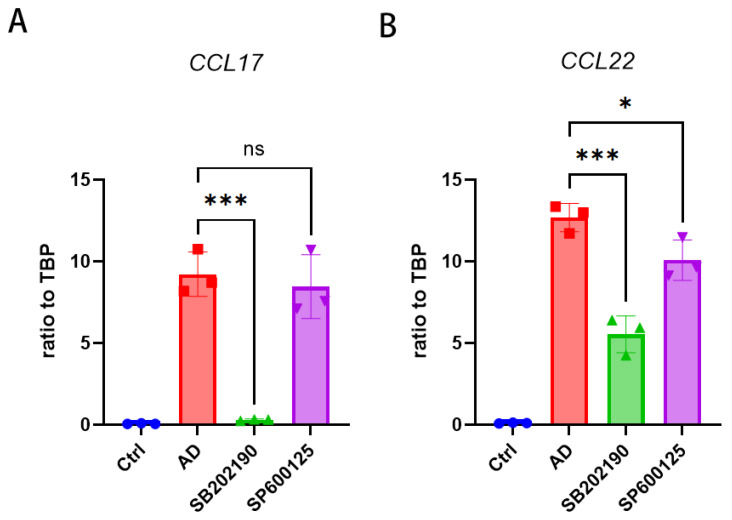
CCL17 and CCL22 expression is promoted by p38 MAPK and JNK in response to TNF-α/IFN- γ. Pre-treatment with SB202190 (5 μM), SP600125 (10 μM) for 30 min. A 20 h application of TNF-α and IFN-γ (10 ng/mL) followed. (**A**) CCL17 mRNA levels. (**B**) CCL22 mRNA levels. A minimum of three experiments were used to calculate the mean ± SEM, *** *p* < 0.001, * *p* < 0.05, ns stands for not significant.

**Figure 5 molecules-28-03507-f005:**
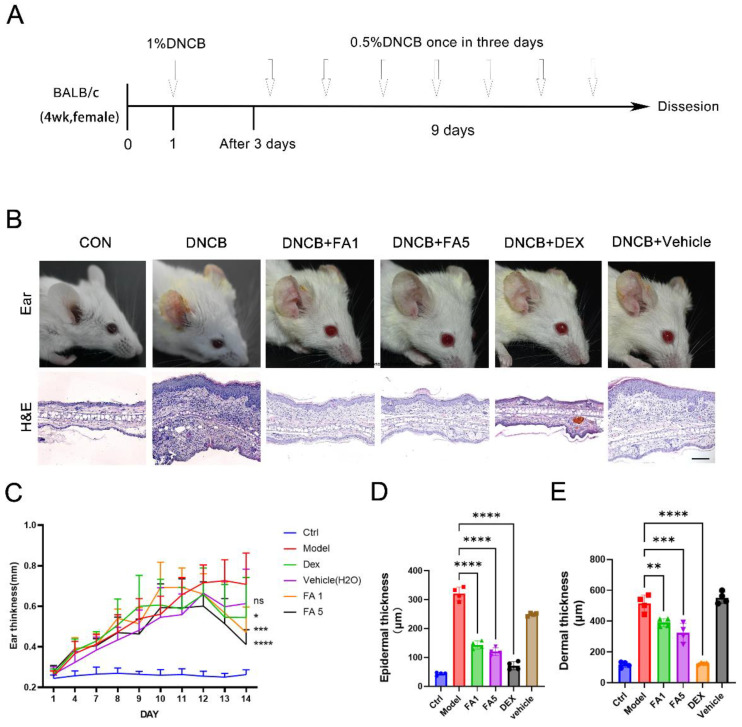
BALB/c mice induced with DNCB show decreased AD-like symptoms and histological changes following treatment with FA. (**A**) Diagram of AD induction. (**B**) Representative images of ear lesions and HE staining of mice on the last day. (**C**) The ear thickness changes in 14 days. (**D**) The changes in epidermis and dermal thickness induced by DNCB (**D**,**E**) (*n* = 4). Scale bar = 100 μm. **** *p* < 0.0001, *** *p* < 0.001, ** *p* < 0.01, * *p* < 0.05, ns means not significant difference.

**Figure 6 molecules-28-03507-f006:**
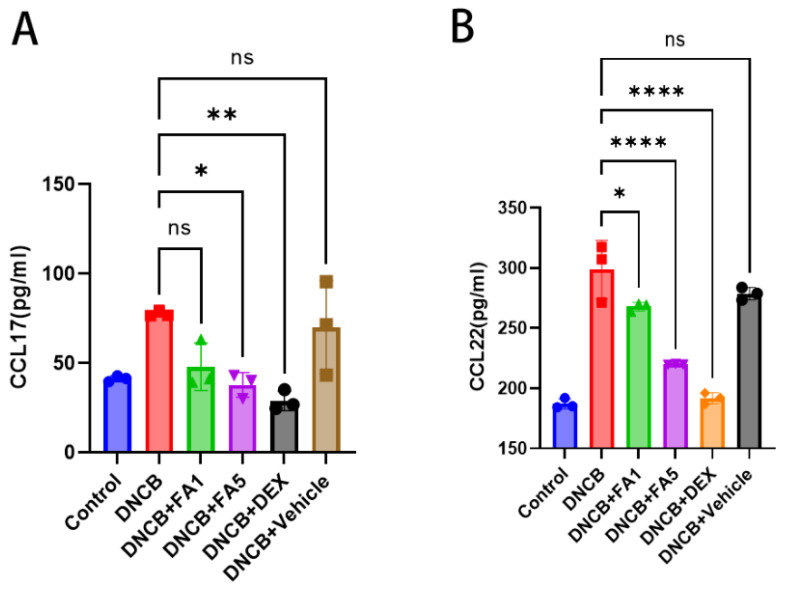
Effects of FA treatment on serum CCL17 and CCL22 levels. The concentrations of CCL17 (**A**) and CCL22 (**B**) in serum were measured using ELISA on day 15 of the study. A minimum of three experiments were used to calculate the mean ± SEM. **** *p* < 0.0001, ** *p* < 0.01, * *p* < 0.05, ns means not significant difference.

**Figure 7 molecules-28-03507-f007:**
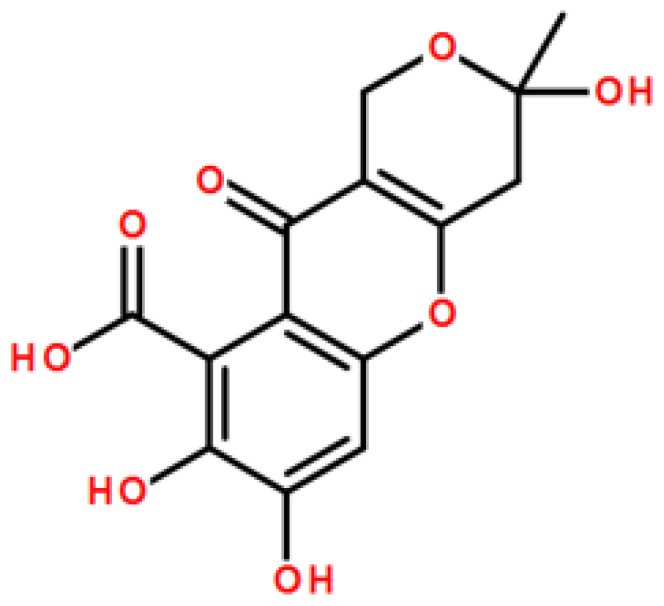
Chemical structure of FA.

## Data Availability

This study did not create or analyze large data sets. Interpretation and reproduction of the data described herein can be provided by the authors upon request.

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
