# Peer review of "Fulvic Acid Attenuates Atopic Dermatitis by Downregulating CCL17/22"

_molecules, 2023, doi:10.3390/molecules28083507_

Round 1

Reviewer 1 Report

In this manuscript titled " Fulvic acid attenuates atopic dermatitis by down-regulating the CCL17/22 ", the author studies the therapeutic effect of FA on AD mice and discussed its potential mechanisms. They find that FA was shown to reduce TARC/CCL17 and MDC/CCL22 expression in HaCaT cells stimulated by TNF-α and IFN-γ by deactivating p38 MAPK and JNK pathways. The research design is appropriate and the result is clearly presented. But I still have some comments or questions

First, could you put the chemical structure of Fulvic acid in the manuscript? so it could be better for the reader to see.

Second, in Fig 3, for the control sample(without TNF-α/IFN-γ treatment), the JNK phosphorylation lever goes up but not goes down after being treated with FA, what's the reason?  

Third, the CCL17 and CCL22 lever does not change significantly by inhibiting the JNK pathway with  SP600125, So does that mean the JNK pathway is not related to the  CCL17 and CCL22 expression in HaCaT cells? 

Forth, I fig 5 D and E, the FA1 and FA5 mean FA 1mg/ml and FA 5mg/ml? make these more clear.

Author Response

I have enriched the article to 4000 words, thank you for your check.

Reviewer 2 Report

The manuscript presents the impact of  fulvic acid (FA) on the level of  CCL17 and CCL22 and  P38 MAPK and JNK in vitro and in vivo models.

Comment:

Please provide more detailed justification for the purpose of this research. The results of the MTT analysis should be statistically verified.

The Authors should present whole membranes for all analyzed proteins, e.g. in Figure 3, panel B, two bands are visible for each sample - please explain this discrepancy compared to panel A

I kindly ask you to expand the discussion and indicate potential opportunities to use the studied  compound in atopic dermatitis

Author Response

I have enriched the article to 4000 words, thank you for your check. Please see the attachment.

Point 1: Please provide more detailed justification for the purpose of this research. The results of the MTT analysis should be statistically verified.

Response 1: I have enriched the purpose of this experiment, interspersed in the introduction and discussion. I did a significance analysis of the cell viability test results and replaced Figure 1.

Point 2: The Authors should present whole membranes for all analyzed proteins, e.g. in Figure 3, panel B, two bands are visible for each sample - please explain this discrepancy compared to panel A

Response 2: The complete bands are shown in Figure 3. The JNK protein appears with two bands due to its antibody properties, as described in the antibody specification.

Point 3:I kindly ask you to expand the discussion and indicate potential opportunities to use the studied  compound in atopic dermatitis.

Response 3:I have expanded the discussion to illustrate the potential opportunities for drug treatment of atopic dermatitis.

Round 2

Reviewer 1 Report

I have never seen any of the author's responses to any of my questions, Please o double-check if you submit that.  the Author Response File is the same file as the revised manuscript.

Author Response

  1. I have added the chemical structure of FA to the Material and Methods section.
  2. In my opinion, this may be due to the error of software quantification. In the control group without TNF α/INF-γ, it is very difficult to identify the naked eye difference in the bands, and there is no significant difference in the quantitative results.
  3. In my opinion, it cannot be concluded that JNK pathway is not related to the expression of CCL17 and CCL22 in HaCaT cells. The expression level of CCL22 can still be inhibited by SP600125 (JNK inhibitor) with significant significance (p<0.05), but P38 MAPK plays a major role in it. If you want me to change this part of the statement I will revise it again. 
  4. I have unified the icon part of the figure and replaced the figure. In the part of Animals and Treatment in the Material and Methods, the grouping of animals and the dosage of drugs have been explained.
  5. I am very sorry that I submitted the wrong manuscript last time. I am very sorry. Please forgive me. Thank you very much for your suggestions on my manuscript.

Reviewer 2 Report

I accept this version

Author Response

Thank you very much for your acceptance.

Round 3

Reviewer 1 Report

The author still doesn't upload your answer to my question in the right place and you put all answers to my question on the author's notes but did not include my question. Please read the Instructions for the Authors. 

Also For Fig 3, I don't see clear does-dependent inhibition of FA when you increase the FA concentration. Do the statistic calculation to see if 200 and 500 have a significant difference or not. 

Round 4

Reviewer 1 Report

The author responded to all my questions and put the file in the right place this time. I still suggest that the author could check the grammar and spelling errors before submitting the final version. 

Author Response

Thank you very much for your advice. I will check my grammar and spelling again before submitting the final version.